# Analysis of Lipid Metabolism in Adipose Tissue and Liver of Chinese Soft-Shelled Turtle *Pelodiscus sinensis* During Hibernation

**DOI:** 10.3390/ijms252212124

**Published:** 2024-11-12

**Authors:** Feng Jin, Yunfei You, Junliang Wan, Huaiyi Zhu, Kou Peng, Zhenying Hu, Qi Zeng, Beijuan Hu, Junhua Wang, Jingjing Duan, Yijiang Hong

**Affiliations:** 1Key Laboratory for Aquatic Germplasm Innovation and Utilization of Jiangxi Province, School of Life Sciences, Nanchang University, Nanchang 330031, China; seanjf2019@163.com (F.J.); pengkou@126.com (K.P.); zengqi9509@hotmail.com (Q.Z.); beijuanhu@ncu.edu.com (B.H.); jhw7454@163.com (J.W.); 2Human Aging Research Institute, Nanchang University, Nanchang 330031, China; youyunfei1027@163.com (Y.Y.); wanjunliang1122@163.com (J.W.); huaiyi_zhu@email.ncu.edu.cn (H.Z.); whozing@aliyun.com (Z.H.)

**Keywords:** hibernation, *Pelodiscus sinensis*, lipid metabolism, metabolic adaptation, cold exposure

## Abstract

Hibernation serves as an energy-conserving strategy that enables animals to withstand harsh environments by reducing their metabolic rate significantly. However, the mechanisms underlying energy adaptation in hibernating ectotherms, such as *Pelodiscus sinensis*, remain contentious. This paper first reports the decrease in lipid levels and the expression of metabolism-related genes in *P. sinensis* during hibernation. The results of physiological and biochemical analysis showed that adipocyte cell size was reduced and liver lipid droplet (LD) contents were decreased during hibernation in *P. sinensis*. Concurrently, serum levels of triglycerides (TGs), total cholesterol (TC), non-esterified fatty acids (NEFAs), high-density lipoprotein cholesterol (HDLC), and low-density lipoprotein cholesterol (LDLC) were diminished (n = 8, *p* < 0.01), while an increase in serum glucose (Glu) (n = 8, *p* < 0.01) was noted among hibernating *P. sinensis*. These observations suggest a shift in energy metabolism during hibernation. To gain insights into the molecular mechanisms, we performed integrated transcriptomic and lipidomic analyses of adipose tissue and livers from summer-active versus overwintering *P. sinensis*, which revealed downregulation of free fatty acids (FFAs), triglycerides (TGs), diglycerides (DGs), and ceramides (Cers) during hibernation. The results of GSEA analysis showed that metabolic pathways associated with lipid metabolism, including glycerolipid metabolism and regulation of lipolysis in adipocytes, were suppressed significantly. Notably, acute cold exposure induced significant downregulation of genes related to lipolysis such as *PNPLA2*, *ABHD5*, *LPL*, *CPT1A*, and *PPARα*. The results indicate that lipolysis is suppressed during hibernation in *P. sinensis*. Collectively, these findings deepen our understanding of survival mechanisms and elucidate the unique energy adaptation strategies employed by hibernating ectotherms. Future research should explore the implications of these findings for the conservation of ectotherms and the applications for artificially inducing hibernation.

## 1. Introduction

Animals can enter a state of hypometabolism or dormancy in response to environmental stressors, such as extreme cold and limited food availability, posing significant challenges to their survival [1]. To cope with the myriad environmental stressors encountered during winter, overwintering amphibians have evolved various behavioral, physiological, biochemical, and molecular adaptations [2,3]. Compared to mammals, hibernation in reptiles has received much less attention [4,5]. Diverse studies have documented a switch from a carbohydrate- to a lipid-based metabolism during hibernation in mammals [6,7,8]. However, the energy supply strategy of ectotherms during hibernation remains elusive.

Hibernation involves complex changes in gene expression and metabolic pathways. High-throughput omics approaches are powerful tools widely employed to investigate animal adaptations to abiotic stresses. For instance, in thirteen-lined ground squirrels, transcriptomic analysis identified succinate dehydrogenase as a key enzyme connecting gene expression to increased ischemic tolerance [9]. Metabolomic studies of thirteen-lined ground squirrels revealed increased levels of free amino acids and β-hydroxybutyrate, which act as alternative energy sources during hibernation [10]. In Tibetan frogs (*Nanorana parkeri*), transcriptome analysis revealed activation of the oxidative stress signaling pathway during hibernation, while metabolome analysis found that increased unsaturated fatty acids maintained membrane fluidity at low temperatures, aiding in ice tolerance [11]. Multi-omics analyses have revealed changes in energy metabolism [12], potential cardio-protective mechanisms [13], mechanisms against muscle atrophy [14], and ischemic tolerance [9] in hibernators.

As an aquaculture animal, the Chinese soft-shelled turtle, *Pelodiscus sinensis*, the growth rate is significantly reduced during hibernation. Previous studies on hibernation in *P. sinensis* have focused on physiological aspects and antioxidant responses [15,16,17,18]. Hibernation is a highly integrative process, making a systems biology approach combined with multi-omics techniques particularly advantageous for exploring the underlying mechanisms. In this study, to obtain deeper insights into the genetic and metabolic alterations associated with hibernation in *P. sinensis*, transcriptomic and lipidomic analyses were conducted on adipose tissue and liver samples collected from non-hibernating (summer) and hibernating (winter) turtles, as these organs play crucial roles in environmental adaptation.

## 2. Results

### 2.1. Histological Observation of Adipose Tissue and Liver

As water temperature decreases to approximately 10 °C in winter, the Chinese soft-shelled turtle exhibits reduced activity and enters a state of hibernation to adapt to the cold (Figure 1A,B). Micro-CT analysis revealed the distribution of adipose tissue in the inguinal region of the turtles, which is highlighted in orange (Figure 1C–E, Appendix A). Morphological analysis of adipose tissue and livers demonstrated significant differences; specifically, hibernating turtles (winter group) exhibited smaller sizes of adipocytes and lipid droplets compared to their autumn, spring, and summer counterparts (Figure 1F–O). The average area of a single adipocyte in adipose tissue was found to be minimal during winter, at 172.2 ± 98.4 μm^2^. In contrast, autumn displayed the maximum average area for a single adipocyte, at 732.6 ± 174.1 μm^2^. The average areas for single adipocytes in spring and summer were measured at 337.1 ± 115.0 μm^2^ and 695.3 ± 152.7 μm^2^, respectively. Notably, the average areas of single adipocytes recorded in autumn, spring, and summer were significantly larger than those observed in winter (*p* < 0.01). During winter, the percentage of lipid droplet area relative to liver area was lowest, at 35.0 ± 4.1%, while it peaked during summer at 51.0 ± 6.8%. The percentages noted for autumn and spring were recorded as 40.7 ± 4.0% and 46.6 ± 7.0%, respectively; remarkably, these percentages during spring and summer were significantly higher than those documented for winter (*p* < 0.01).

### 2.2. The Changes in Serum Biochemical Indicators

To evaluate the changes in lipid-metabolism-related indicators in turtle serum, the contents of triglyceride (TG), total cholesterol (TC), non-esterified fatty acid (NEFA), high-density lipoprotein cholesterol (HDLC), low-density lipoprotein cholesterol (LDLC), and glucose (Glu) were evaluated. Compared to the winter group, there were no significant fluctuations in serum lipid levels in autumn, spring, or summer (Figure 1P–T). Notably, the winter group had higher levels of serum glucose than autumn and spring (Figure 1U).

### 2.3. Identification and Functional Enrichment Analysis of Differentially Accumulated Lipids (DALs)

Winter and summer represent the characteristic phases of hibernation and non-hibernation states in *Pelodiscus sinensis*, respectively. To further investigate this phenomenon, mass-spectrometry-based lipidome testing was employed to assess variations in lipid composition and distribution in adipose tissue and livers during non-hibernation (summer group) and hibernation (winter group). Principal component analysis (PCA) revealed that samples between groups were dispersed while those within groups were clustered, indicating robust replication within groups and significant differences between them (Figure 2A,B). A total of 459 differentially accumulated lipids (DALs) were identified in adipose tissue, comprising 281 downregulated and 178 upregulated lipids (Figure 2C). The livers exhibited a total of 702 DALs, including 629 downregulated and 73 upregulated lipids (Figure 2D). All significantly altered lipid species were visualized using a bubble map (Figure 2E,F). In adipose tissue, 231 glycerophospholipids were downregulated during hibernation, along with a decrease in 10 ceramides and 15 glycerides. In the liver, there was a downregulation of 344 types of glycerides during hibernation, as well as a reduction in 208 types of glycerophospholipids and 21 types of ceramides. Additionally, KEGG pathway analysis highlighted significant modifications in the adipocytokine signaling pathway as well as the AGE-RAGE signaling pathway related to diabetic complications and cardiomyopathy in both adipose tissue and liver tissues (Figure 2G,H). Further analysis indicated that the most substantial decrease occurred in fatty acid content, specifically those with carbon chain lengths ranging from C19 to C26 (Figure 3A), alongside decreases in triglycerides and diglycerides from adipose tissue samples (Figure 3B,C). In the liver, reductions were noted for fatty acids and diglycerides during hibernation; however, the triglyceride content exhibited the most pronounced decline overall (Figure 3D,E and Appendix A). Ceramides, a crucial class of sphingolipids important for cellular signaling, were significantly reduced in both liver (Figure 3F) and adipose tissues (Figure 3G) during hibernation.

### 2.4. Identification and Functional Enrichment Analysis of Differentially Expressed Genes (DEGs)

RNA-seq analysis was conducted to elucidate the adaptive mechanisms employed by *P. sinensis* during hibernation. Each point in the PCA plot represented a sample, with samples from the same group clustering together while different groups exhibited a trend of separation (Figure 4A,B). Significant differences were identified among 4588 differentially expressed genes (DEGs) in adipose tissues, comprising 2466 downregulated and 2122 upregulated genes (Figure 4C). Similarly, a total of 1876 DEGs were detected in the liver, including 1064 downregulated and 812 upregulated genes (Figure 4D). KEGG pathway enrichment analysis revealed associations with multiple pathways; notably, hibernation had the most significant impact on metabolic pathways in the Chinese soft-shelled turtle. Additionally affected pathways included thyroid hormone signaling, endocrine resistance, and amino acid metabolism (Figure 4E,F). To validate the RNA-seq findings, ten DEGs were randomly selected for qRT-PCR analysis. The qRT-PCR results demonstrated that mRNA expression levels of *TPPP3*, *NAP1L4*, *TANK*, *MAOA*, and *RGS4* genes significantly decreased, as observed in the transcriptome data; conversely, *IL1R1*, *DGAT2*, *VGLL1*, *PRPF38A*, and *SORL1* genes showed upregulation consistent with transcriptomic findings during hibernation compared to non-hibernation states (Appendix A).

A total of 349 KEGG pathways enriched by differentially expressed genes (DEGs) and 46 KEGG pathways enriched by differentially accumulated lipids (DALs) were comprehensively analyzed in adipose tissue, revealing simultaneous enrichment of glycerolipid metabolism and regulation of lipolysis in adipocytes (Figure 5A). Subsequently, to ascertain whether these two pathways were upregulated or downregulated, gene set enrichment analysis (GSEA) was performed on the differential genes associated with both pathways. The normalized enrichment score (NES) for glycerolipid metabolism was -0.808 (Figure 5B), while the NES for regulation of lipolysis in adipocytes was −0.902 (Figure 5C). Additionally, a comprehensive analysis of the 342 KEGG pathways enriched by DEGs and the 106 KEGG pathways enriched by DALs in liver tissues revealed an NES of −1.166 for glycerolipid metabolism (Figure 5E), and an NES of −1.283 for regulation of lipolysis in adipocytes (Figure 5F). These GSEA results indicated suppression of these metabolic pathways during hibernation in both adipose tissue and livers.

### 2.5. The Expression of Genes Involved in Lipid Metabolism Under Acute Cold Exposure

The expression of lipolysis-related genes (*PNPLA2*, *ABHD5*, *LPL*, *CPT1A*, and *PPARα*) in *Pelodiscus sinensis* was reduced in both adipose tissue (Figure 6A) and livers (Figure 6B) during hibernation. To investigate whether physical low-temperature stimulation induces metabolic inhibition in *P. sinensis*, the turtles were subjected to acute cold conditions (Figure 6C). Acute exposure to 10 °C for 48 h resulted in decreased mRNA expression levels of lipolysis-related genes (*PNPLA2*, *ABHD5*, *LPL*, *CPT1A*, and *PPARα*) in adipose tissue (Figure 6D). Similarly, the mRNA expression levels of these genes in the livers were also diminished, indicating suppressed lipid metabolism (Figure 6E).

## 3. Discussion

### 3.1. Pelodiscus sinensis Does Not Mainly Rely on Fat Store Strategy to Survive Hibernation

During hibernation, white adipose tissue (WAT), the primary energy storage organ, undergoes shrinkage as mammals rely on its degradation for energy [19]. Studies on Ursus arctos have demonstrated smaller adipocyte sizes in summer captures compared to winter [20]. In contrast, we observed a reduction in the sizes of individual adipocytes in the adipose tissue of *Pelodiscus sinensis* during hibernation. The liver, a crucial metabolic organ, stores excess energy as triglycerides (TGs) within lipid droplets (LDs) and releases free fatty acids during periods of energy deficit. Lipid droplet contents in the livers of squirrels have been shown to increase significantly during hibernation [21]. Contrary to histological observations in mammals, both adipocyte size and liver LD abundance are greater during non-hibernation; additionally, the proportion of adipose tissue and liver mass relative to body weight increases. For the Mississippi map turtle (*Graptemys pseudogeographica*), liver lipid levels were highest during the reproductive period and lowest prior to hibernation [22]. The eastern box turtle (*Terrapene carolina*) exhibits seasonal changes in body weight; however, lipid storage and utilization remain unaltered during hibernation [23]. Similar to findings observed in *P. sinensis*, turtles undergoing hibernation show reductions in hepatocyte LDs and TG content, suggesting limited fat reserves throughout this period [24].

Elevated serum levels of triglycerides (TGs), total cholesterol (TC), non-esterified fatty acids (NEFAs), high-density lipoprotein cholesterol (HDLC), and low-density lipoprotein cholesterol (LDLC) serve as critical markers for robust lipid metabolism. During hibernation, increased blood lipid levels indicate heightened lipolysis from adipose tissue in *Dromiciops gliroides*, *Ursus arctos*, *Nyctereutes procyonoides albus*, and *Erinaceus europaeus* [25,26,27,28]. This phenomenon likely arises from mammals relying on stored lipids to sustain energy and essential physiological functions during hibernation. Previous research has indicated reduced serum biochemistry parameter values related to lipolysis during hibernation in ectotherms such as reptiles and amphibians, including the radiated tortoise [29], desert tortoise *Gopherus agassizii* [30], yellow-marginated box turtle *Cuora flavomarginata* [31] and Chinese soft-shelled turtle *Pelodiscus sinensis* [32]. The levels of TG, TC, NEFA, HDLC, and LDLC in serum remained low during hibernation in our study, which aligns with those observed in these amphibians and reptiles but contrasts with findings in mammals. This may suggest that ectotherms are not as metabolically active in terms of lipid metabolism during hibernation, indicating that the Chinese soft-shelled turtle does not primarily rely on fat reserves for energy during this period. In conclusion, mammals and ectotherms employ distinct strategies for lipid metabolism to cope with cold environments; specifically, ectotherms like *P. sinensis* seem not to depend on extensive fat reserves for survival during hibernation.

### 3.2. Lipid Metabolism Is Suppressed During Hibernation in Pelodiscus sinensis

Our findings elucidate cold-induced lipid dynamics and gene expression patterns in adipose tissue and livers, providing lipidomic and transcriptional signatures in response to cold exposure. During hibernation, mammals mobilize fat for energy while simultaneously enhancing fat storage to endure prolonged cold periods, predominantly relying on lipid metabolism for energy, with fatty acids from white adipose tissue serving as the primary fuel source [33,34]. Lipoprotein lipase (*LPL*) catalyzes the hydrolysis of triacylglycerols present in chylomicrons within adipose tissue, the heart, and skeletal muscle, resulting in the production of fatty acids (FAs) and glycerol [35]. In contrast, the oxidation of fatty acids is regulated by the rate-limiting enzymes *CPT1A* in the liver and *CPT1B* in both the heart and skeletal muscle [36]. The upregulation of lipid-catabolism-related genes leads to significant increases in triglyceride and fatty acid concentrations during hibernation compared to non-hibernation periods in mammals [21,37]. In contrast, studies on ectotherms have shown that the expression of genes related to glycolysis, lipid metabolism, and amino acid metabolism is downregulated in hibernating Chinese alligators, indicating a suppression of energy metabolism [38]. Similar results have been observed in Tibetan frogs (*Nanorana parkeri*) and Asiatic toads (*Bufo gargarizans*) [11,39]. *P. sinensis* experiences a decrease in triglycerides, diglycerides, and fatty acids during hibernation due to temperature-induced downregulation of lipolysis-related genes (*PNPLA2*, *ABHD5*, *LPL*, *CPT1A*, and *PPARα*) as well as pathways involved in glycerolipid metabolism and regulation of lipolysis in adipocytes, contrasting with the typical increase seen in hibernating mammals. These results indicate differences in gene expression patterns between mammals and amphibians during hibernation, suggesting that amphibians possess unique molecular regulatory mechanisms. Such variation likely arises from differences in thermoregulatory mechanisms between the two groups; however, further research is warranted to fully elucidate these mechanisms.

### 3.3. Low Temperature Stimulates Lipid Metabolism Inhibition in Pelodiscus sinensis

Temperature plays a pivotal role in regulating lipid metabolism in ectotherms such as *P. sinensis*. *PNPLA2* acts as a catalytic enzyme in the initial step of triglyceride hydrolysis and serves as a key protein in the regulation of lipid metabolism [40,41]. *CGI-58* (also known as *ABHD5*) can directly interact with the Patatin domain of *PNPLA*, which is necessary for *PNPLA* to respond to lipolysis signals and exhibit full enzyme activity [42,43]. An increase in lipid catabolism is primarily mediated by *PPARα*, which transcriptionally regulates nearly all enzymes involved in mitochondrial uptake and oxidative breakdown of fatty acids, making it an essential regulator of hibernation-related lipid metabolism in mammals [44]. Low temperatures induce hibernation in *P. sinensis*, accompanied by the downregulation of lipolysis-related genes (*PNPLA2*, *ABHD5*, *LPL*, *CPT1A*, and *PPARα*), underscoring the critical influence of temperature in modulating lipid metabolism during this period. While exposure to cold has been shown to enhance lipid metabolism in mice, our findings indicate the opposite effect in Chinese soft-shelled turtles [12,45]. Even short-term exposure to low temperatures can inhibit lipid metabolism in these turtles. This adaptation to cold stress may reflect an evolutionary mechanism that enables ectotherms like *P. sinensis* to conserve energy and survive harsh conditions. However, the molecular mechanism of the *P. sinensis* sensing temperature signal warrants further attention.

## 4. Materials and Methods

### 4.1. Turtles and Sample Collection

A total of 32 Chinese soft-shelled turtles were captured in a culture pond in Nanfeng County (GPS coordinates N 116.62 E 27.05), Jiangxi, China. Eight turtles were collected in autumn (early November, water temperature approximately 24 °C), eight in winter (early January, water temperature approximately 10 °C), eight in spring (early April, water temperature approximately 20 °C), and eight in summer (late August, water temperature approximately 32 °C). Before the sample collection, the health status of the turtles was assessed as normal without any abnormalities. Turtles with similar body weights ranging from 200.0 to 300.0 g were selected to ensure uniformity among groups. The turtles were anesthetized by 100 mg L-1 MS-222 (Sigma, St. Louis, MO, USA) for 2 to 4 min and subsequently euthanized via cervical bleeding with a sterile dissection kit. The blood samples were collected and transferred to an EDTA tube. After resting at room temperature for 30 min, the blood samples were centrifuged at 3000× *g* for 30 min at room temperature to obtain serum, which was stored at -80 °C for further analysis. A portion of adipose tissue from the inguinal region and liver was immediately sampled after euthanasia and preserved in liquid nitrogen for lipidomic and transcriptomic analysis. Adipose tissue and liver samples designated for morphological analysis were immersed in 4% PFA for at least 24 h. All procedures involving turtle handling complied with regulations set forth by the Animal Center of Nanchang University, China; sampling protocols received approval from the School of Life Sciences at Nanchang University under approval ID SYXK(GAN)2021-0004. All efforts were made to minimize the animals’ suffering.

### 4.2. Experimental Turtles and Sample Collection for Cold Stress Analysis

Twelve juvenile turtles were housed in two water tanks (dimensions: 70 cm × 50 cm × 40 cm, water depth: approximately 10 cm), with 6 turtles per tank, for a week-long acclimation period. The temperature was maintained at 32 ± 0.5 °C under a photoperiod of 12 h light and 12 h dark. Half of the water in each tank was replaced daily with isothermal, air-saturated tap water. Following the acclimation period, all turtles underwent a fasting phase lasting 48 h. Turtles designated for the cold stress group (n = 6) were randomly selected from the tanks. Subsequently, the water temperature was rapidly decreased from 32 °C to 10 °C by 1 °C per hour using a cooling apparatus and maintained at this temperature for an additional 48 h, during which all turtles remained submerged in cold water. Afterward, adipose tissue and liver samples were promptly excised from all animals and flash-frozen in liquid nitrogen for further analysis.

### 4.3. Micro-CT Analysis

A Micro CT imaging system (Quantum GX, PerkinElmer, Boston, MA, USA) was used for adipose tissue analysis, with the parameters settings presented in Table 1, as we published before [46]. Before usage, the instrument was preheated for 15 min to ensure stability. The surface of the turtle was dried with a towel before placing it into the sample tank. Transmission imaging was directly performed through an X-ray for the initial imaging. The position of each turtle was adjusted manually or via adjustment buttons on the instrument for 3D scanning at three angles (90°, 180°, 270°) with a laser source check. Initial imaging of each living turtle via micro-CT indicated the location of adipose tissue. Then, 3D images and videos of each scanned specimen were generated and further analyzed using the equipped software.

### 4.4. Histology

Turtle adipose tissues and livers were fixed in 4% formalin buffer (Servicebio, Wuhan, Hubei, China) and O.C.T. compound (Servicebio, Wuhan, Hubei, China), and cryosections (8 μm) were prepared. According to standard protocols, the adipose tissue sections were stained with hematoxylin and eosin (H&E) (Servicebio, Wuhan, Hubei, China), and livers were stained with oil red O dye (Servicebio, Wuhan, Hubei, China). Six slices were randomly selected from each group, and five visual fields were randomly selected from each slice. The size of fat cells in the tissue was observed using an optical microscope (ECLIPSE Ni, Nikon, Tokyo, Japan) and measured using NIS-Elements Viewer software (v4.2.0, Nikon, Tokyo, Japan). The deposition of hepatic lipid droplets in the tissue was observed using an optical microscope (ECLIPSE Ni, Nikon, Tokyo, Japan). Images were obtained in a blinded manner and analyzed using ImageJ software (v1.8.0, NIH, Bethesda, MD, USA).

### 4.5. Biochemical Analysis of Serum Samples

Serum levels of triglycerides (TGs), total cholesterol (TC), non-esterified fatty acids (NEFAs), high-density lipoprotein cholesterol (HDLC), and low-density lipoprotein cholesterol (LDLC) were analyzed using detection kits according to the manufacturers’ instructions (Jiancheng, Nanjing, China).

### 4.6. UPLC–MS/MS Analysis

Adipose tissue and livers from non-hibernation and hibernation turtles were used for lipidomics analysis. We subjected 50 mg of each tissue sample to liquid extraction. Ultra-performance liquid chromatography (UPLC) and tandem mass spectrometry (MS/MS) analysis were performed at Metware Biotechnology Co., Ltd. (Wuhan, China).

### 4.7. Differentially Accumulated Lipids (DALs) and Enrichment Analysis

The lipidomic data were analyzed by principal component analysis (PCA) using the ropes package in R (v2.0, Jasmine Mountain, NJ, USA). Both ropes and SciPy were used to calculate the variable importance in the projection (VIP) values, while stats (R package) and SciPy (Python package) were used to calculate the FDR using a paired-samples *t*-test, with FDR < 0.05 and VIP > 1 indicating significant differential lipids. KEGG pathway enrichment, correlation, and cluster analyses were performed for differential lipids using SciPy (Python, v2.0, PSF, Nanakht, NY, USA). Fisher’s exact test was used to identify significantly enriched pathways (FDR < 0.05).

### 4.8. RNA Extraction and RNA-Seq Analysis

Total RNA from adipose tissues and livers (20 mg) was extracted using the RNAprep Pure Tissue kit (DP431, Tiangen, Beijing, China). Nucleic acids were released via the addition of lysis buffer to the tissue. Following liquid-phase stratification, RNA was adsorbed onto an adsorption column. The OD260/OD280 was in the normal range of 1.8–2.0, and the integrity of the RNA was assessed on 1.0% agarose gel. RNA quality was assessed using a Nano Photometer spectrophotometer (IMPLEN, CA, USA). Poly(A) mRNA was enriched using magnetic beads with oligo(dT). The mRNA was fragmented randomly, and first-strand cDNA was synthesized using the M-MuLV reverse transcriptase system. RNase H was then used to degrade the RNA strand, and second-strand cDNA was synthesized using DNA polymerase. The double-stranded cDNAs were ligated to sequencing adapters. cDNAs (~200 bp) were size-selected using AMPure XP beads. Following amplification and purification, cDNA libraries were prepared and sequenced on the Illumina Novaseq6000 system. Illumina RNA-Seq was conducted by Metware Biotechnology Co., Ltd. (Wuhan, China).

### 4.9. Differentially Expressed Genes (DEGs) and Enrichment Analysis

Transcriptomic data were analyzed using the Metware Cloud Platform (v2.0, https://cloud.metware.cn, accessed on 6 November 2024). RSEM software (v1.3.1) was utilized to quantitatively analyze transcript expression based on transcripts per million (TPM) values. DESeq2 was employed to assess differences in expression between samples, with a *P*-adjust (FDR < 0.05 and |log2FC| > 1.0) indicating significant DEGs. KOBAS (http://bioinfo.org/kobas, accessed on 6 November 2024) was utilized for the Kyoto Encyclopedia of Genes and Genomes (KEGG) pathway enrichment analysis. Fisher’s exact test was employed to identify significantly enriched KEGG pathways (FDR < 0.05). R software (v2.0) was used for cluster analysis, heatmap visualization, and Pearson correlation assessment.

### 4.10. Quantitative PCR

Purified RNA (1 μg per sample) was reverse transcribed to first-strand cDNA using a cDNA Reverse Transcription Kit (Prime Script^TM^ RT Master Mix, Takara, Kusatsu, Japan) following the manufacturer’s instructions. The primer sequences are provided in Appendix A. RT-qPCR was conducted in triplicate using the gene expression assay (Applied Biosystems, Carlsbad, CA, USA) on an Applied Biosystems Fast 7500 machine (Thermo Fisher Scientific, Waltham, MA, USA). β-actin served as internal controls for normalization and relative quantification.

### 4.11. Statistical Analysis

All data are presented as the mean ± standard error of the mean (SEM). Shapiro–Wilk’s test was used to test for differences in each parameter among all the groups. Homogeneity of variance was assessed by Levene’s test, and Tukey’s test was performed on the samples to identify significant differences between the two groups. All statistical analyses were conducted using the SPSS 22.0 software (IBM, Armonk, NY, USA). Statistical significance was set at *p* < 0.05. We denoted significance levels with * and **, representing *p*-values of less than 0.05 and 0.01 between groups, respectively. Bar graphs were made using GraphPad Prism 8.0 software (GraphPad Software, La Jolla, CA, USA).

## 5. Conclusions

Before entering hibernation, mammals accumulate substantial fat reserves to sustain their energy supply during this period. However, our findings suggest that Chinese soft-shelled turtles employ a different strategy during hibernation. Unlike mammals, *Pelodiscus sinensis* does not rely on the accumulation of large lipid stores during hibernation. Instead, it suppresses lipid metabolism to adapt to hibernation conditions. Specifically, as environmental temperature drops during hibernation, there is a significant reduction in fatty acid β-oxidation and downregulation of genes associated with fat mobilization, accompanied by decreases in triglyceride, diglyceride, and free fatty acid contents in Chinese soft-shelled turtles. In conclusion, our research not only advances the field of hibernation biology but also highlights the intricate and diverse strategies employed by different species to survive harsh environmental conditions. Furthermore, the insights gained from this study have the potential to influence future research and conservation efforts targeting ectothermic species that undergo hibernation.

## Figures and Tables

**Figure 1 ijms-25-12124-f001:**
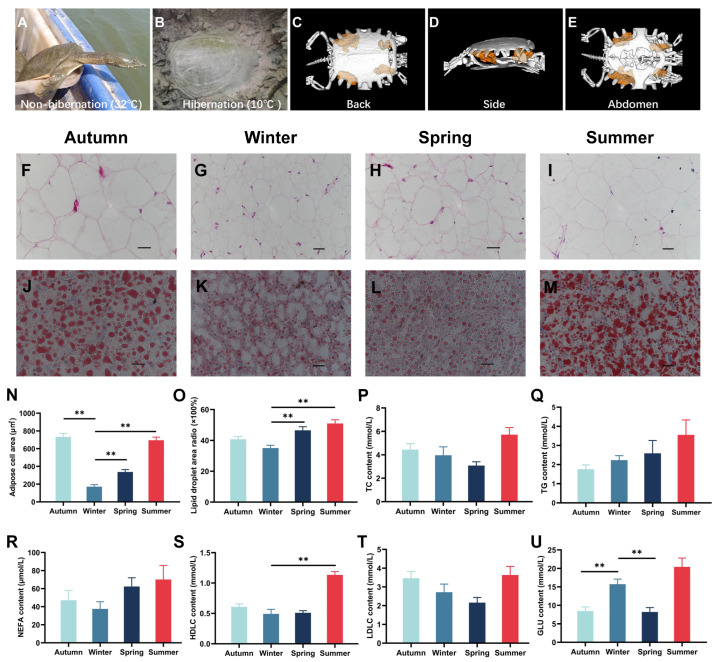
Fat masses in adipose tissue and livers, and lipid indexes in serum of *Pelodiscus sinensis* in four seasons. (**A**,**B**) Images of *P. sinensis* in non-hibernation (**A**) and hibernation (**B**) states under different environmental temperatures. (**C**–**E**) The 3D images from X-ray microcomputed tomography (micro-CT) scanning of *P. sinensis*. For more detail from the micro-CT 3D images, see Appendix A. (**F**–**I**) Representative hematoxylin and eosin (H&E) staining images of adipose tissue from inguinal region. Scale bars, 10 μm. (**J**–**M**) Representative oil red O staining images of liver. Scale bars, 10 μm. (**N**) Quantification of the sizes of individual adipocytes (n = 30). (**O**) Percentage of lipid droplet area in the total sample area (n = 30). (**P**–**U**) The levels of biochemical indexes related to lipid metabolism in serum. TG: triglyceride; TC: total cholesterol; NEFA: non-esterified fatty acid; HDLC: high-density lipoprotein cholesterol; LDLC: low-density lipoprotein cholesterol; Glu: glucose (n = 8). **, *p* < 0.01 (relative to winter).

**Figure 2 ijms-25-12124-f002:**
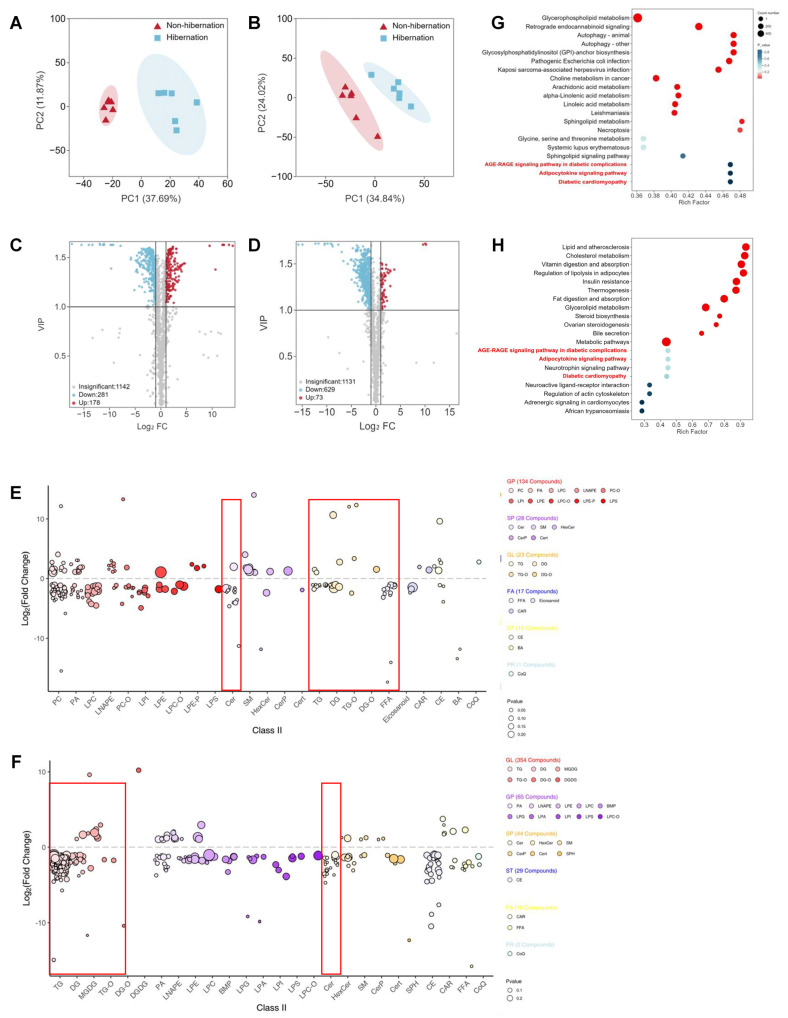
Lipidomic analysis of differentially accumulated lipids (DALs) during non-hibernation and hibernation. (**A**,**B**) Principal component analysis (PCA) score plots of adipose tissue (**A**) and liver (**B**) lipid profiles. (**C**,**D**) Volcano map of lipid detection results in adipose tissue (**C**) and livers (**D**). (**E**,**F**) The fold change in concentration of all quantified lipid species between non-hibernation and hibernation groups in adipose tissue (**E**) and livers (**F**). Each dot represents a lipid species, and the dot size indicates significance. The different lipid classes are color-coded. (**G**,**H**) KEGG functional pathway annotation of differentially accumulated lipids in adipose tissue (**G**) and livers (**H**).

**Figure 3 ijms-25-12124-f003:**
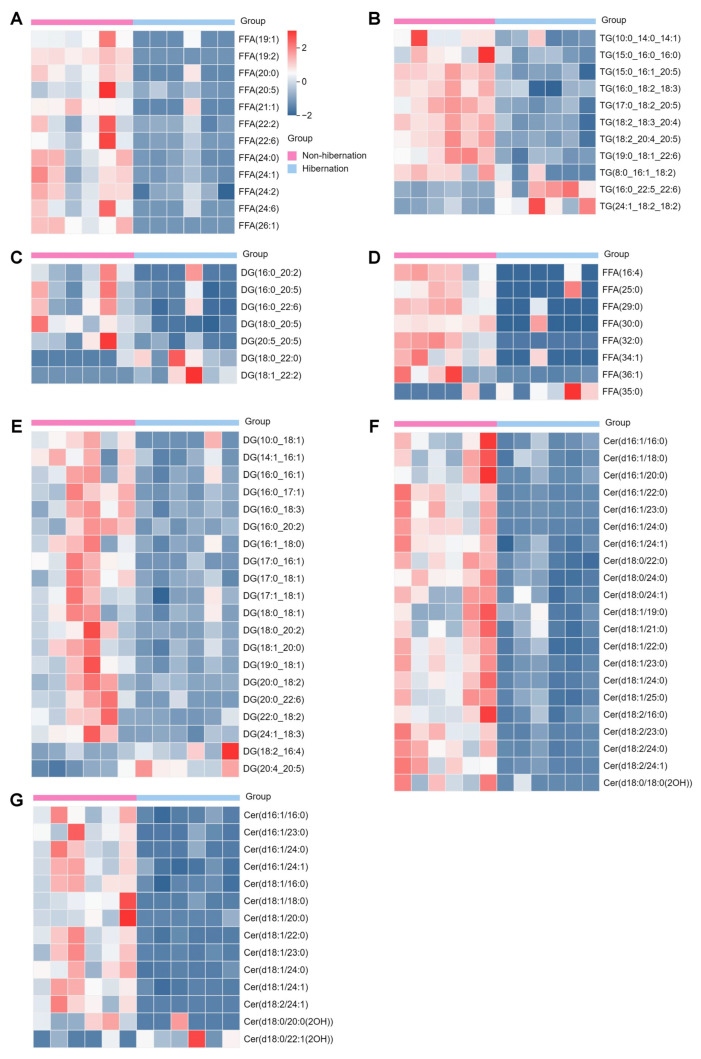
Distinct lipid profile of non-hibernating and hibernating *P. sinensis*. (**A**–**D**) Heatmap analysis of FFA species (**A**), DG species (**B**), TG species (**C**), and Cer species (**D**) in adipose tissue. (**E**–**G**) Heatmap analysis of FFA species (**E**), DG species (**F**), and Cer species (**G**) in livers. FFA: free fatty acid; DG: diacylglycerol; Cer: ceramide.

**Figure 4 ijms-25-12124-f004:**
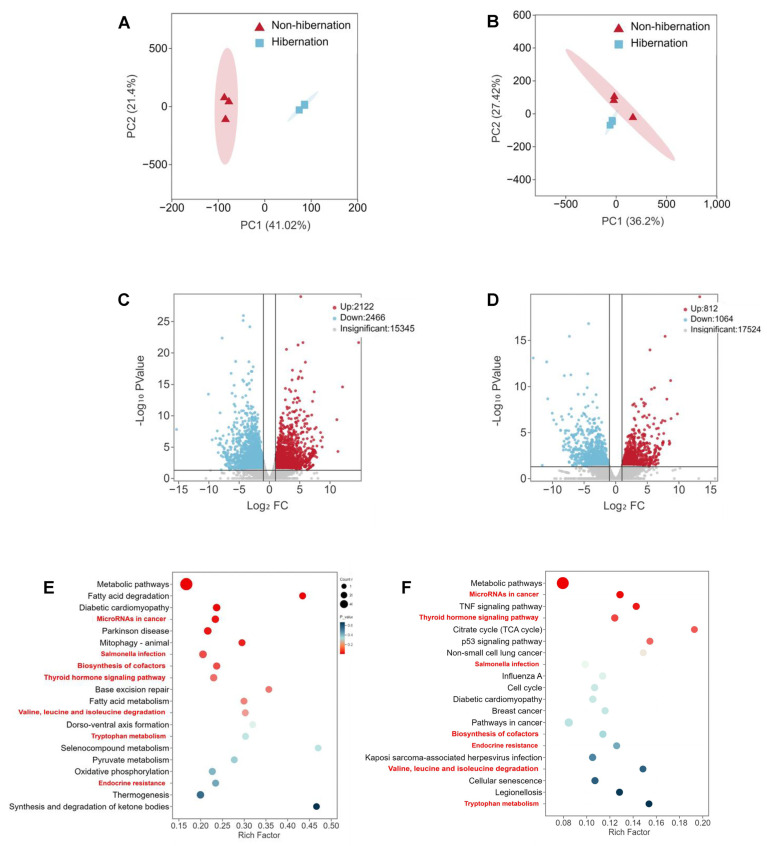
RNA-seq analysis of DEGs during non-hibernation and hibernation. (**A**,**B**) Principal component analysis (PCA) score plot of adipose tissue (**A**) and liver (**B**) transcription profiles. (**C**,**D**) Volcano map of gene detection results in adipose tissue (**C**) and livers (**D**). (**E**,**F**) KEGG enrichment analysis of differentially expressed genes in adipose tissue (**E**) and livers (**F**).

**Figure 5 ijms-25-12124-f005:**
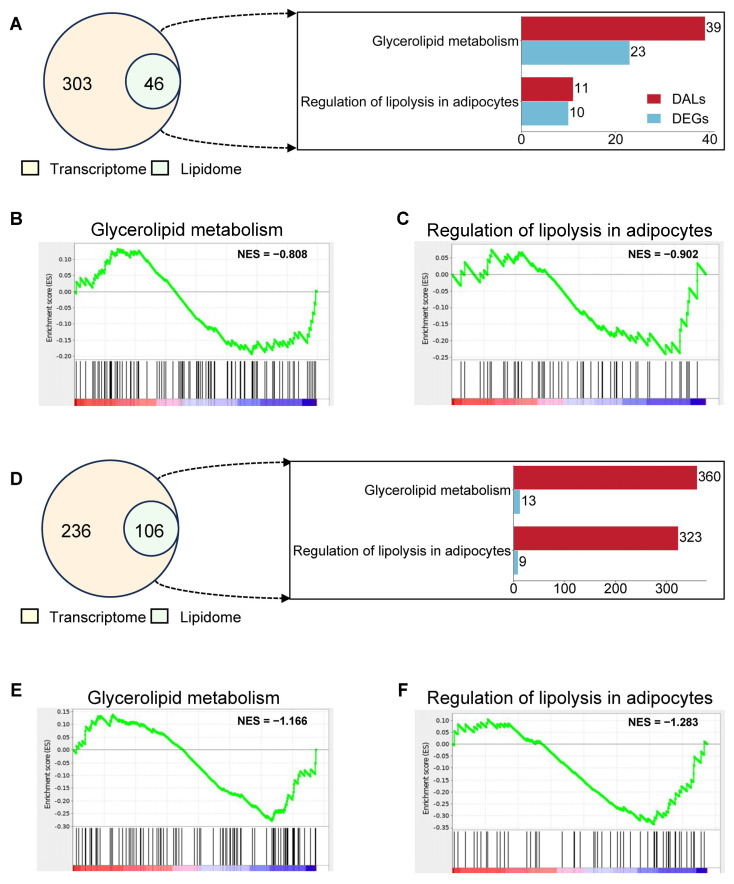
Lipid-metabolism-related pathways are suppressed during hibernation. (**A**) The Venn diagram shows the overlapping KEGG pathways of transcriptome and lipidome from adipose tissue. Glycerolipid metabolism and regulation of lipolysis in adipocyte pathways are the overlapping KEGG pathways related to lipid metabolism. (**B**,**C**) Gene set enrichment analysis (GSEA) of glycerolipid metabolism (**B**) and regulation of lipolysis in adipocytes (**C**) found in gene sets from adipose tissue. (**D**) The Venn diagram shows the overlapping KEGG pathways of the transcriptome and lipidome from livers. Glycerolipid metabolism and regulation of lipolysis in adipocyte pathways are the overlapping KEGG pathways related to lipid metabolism. (**E**,**F**) GSEA of glycerolipid metabolism (**E**) and regulation of lipolysis in adipocytes (**F**) found in gene sets from livers.

**Figure 6 ijms-25-12124-f006:**
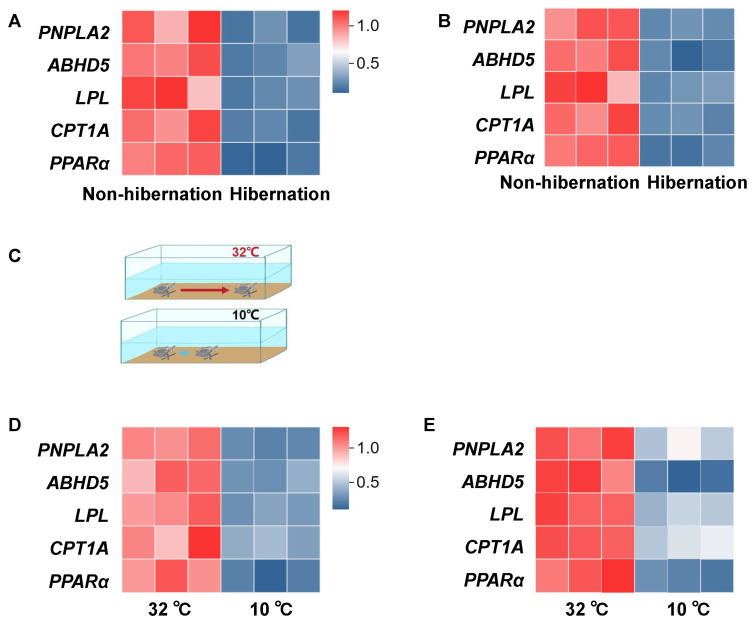
Low-temperature exposure in *P. sinensis*. (**A**,**B**) The relative mRNA expression of lipolysis-related genes (*PNPLA2*, *ABHD5*, *LPL*, *CPT1A*, and *PPARα*) in adipose tissue (**A**) and livers (**B**) (n = 3) during non-hibernation and hibernation. (**C**) Schematic of crawling distance measurement of *P. sinensis* under cold exposure. (**D**,**E**) The relative mRNA expression of lipolysis-related genes (*PNPLA2*, *ABHD5*, *LPL*, *CPT1A*, and *PPARα*) in adipose tissue (**D**) and livers (**E**) from 32 °C group and 10 °C group.

**Table 1 ijms-25-12124-t001:** Parameters of micro-CT for *Pelodiscus sinensis* scanning.

Parameter	Value
Voltage (V)	90
Current (V)	88
Acquisition (mm)	72
Recon (mm)	72
Scan Mode	High Speed
Period (s)	8

## Data Availability

The original contributions presented in the study are included in the article/Appendix A, further inquiries can be directed to the corresponding author due to privacy reasons.

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
