# Peer review of "Analysis of Lipid Metabolism in Adipose Tissue and Liver of Chinese Soft-Shelled Turtle *Pelodiscus sinensis* During Hibernation"

_ijms, 2024, doi:10.3390/ijms252212124_

Round 1
Reviewer 1 Report
Comments and Suggestions for Authors
See the attached file.

Author Response
Dear Reviewer,
We would like to express our heartfelt gratitude for your valuable suggestions and comments. We have included our responses in the attached document. Please see the attachment.

Reviewer 2 Report
Comments and Suggestions for Authors
Analysis of Lipid Metabolism in Adipose Tissue and Liver of Chinese Soft-Shelled Turtle Pelodiscus sinensis During Hibernation
Dear Authors,
The manuscript is very interesting, and very well prepared. Minor revision is required from point of view edition of text, order of sections in manuscript, more precise description of Statistical analysis subsection and some corrections in References section.
Below I add some suggestions helpful in this process:
Line 2
Maybe better is to use capital letters in case of title:
Analysis of Lipid Metabolism in Adipose Tissue and Liver of Chinese Soft-Shelled Turtle Pelodiscus sinensis During Hibernation
Line 26
In text of manuscript is P. sinensis, lack of italics.
Line 51
Space required after tolerance and before [9].
Line 61
Order of sections must be changed, Materials and Methods after Discussion.
11. Introduction.
22. Results (numeration of subsections must be also adapted).
33. Discussion.
44. Materials and Methods (numeration of subsections must be also adapted).
55. Conclusions.
Line 163
Information about normality of distribution in case of each seasonal treatment required (Shapiro-Wilk’s test) and homogeneity of variance conducted by Levene’s test.
Line 165
In case of significant differences post-hoc test is required. Tukey’s test is the best choice in this case.
Lines 167, 168, 184, 188
p-value instead of P-value must be used (sample from population).
Line 206
Description in form of sentence can be added for asterixis applied in charts N-U.
I.e.:** Differences between seasons significant at p<0.01.
Line 251
P. sinensis
Lines 413-505
References section
Doi links required on the end of each reference.
Number of volume must be also italicized.
Lines 416, 417, 427, 463, 499 and 502
Abbreviations in case of Journal’s name are required.
Author Response

(The authors gave the same response as above.)
